# Ferritin and C-reactive protein are predictive biomarkers of mortality and macrophage activation syndrome in adult onset Still's disease. Analysis of the multicentre *Gruppo Italiano di Ricerca in Reumatologia Clinica e Sperimentale* (GIRRCS) cohort

**Paola Di Benedetto**[1☯], **Paola Cipriani**[2☯], **Daniela Iacono**[3], **Ilenia Pantano**[3], **Francesco Caso**[4], **Giacomo Emmi**[5], **Rosa Daniela Grembiale**[6], **Francesco Paolo Cantatore**[7], **Fabiola Atzeni**[8], **Federico Perosa**[9], **Raffaele Scarpa**[4], **Giuliana Guggino**[10], **Francesco Ciccia**[3], **Roberto Giacomelli**[2‡]*, **Piero Ruscitti**[2‡]

**1** Clinical Pathology Unit, Department of Biotechnological and Applied Clinical Sciences, University of L'Aquila, L'Aquila, Italy, **2** Rheumatology Unit, Department of Biotechnological and Applied Clinical Sciences, University of L'Aquila, L'Aquila, Italy, **3** Rheumatology Section, Department of Clinical and Precision Medicine, University of Campania "Luigi Vanvitelli", Naples, Italy, **4** Rheumatology Unit, Department of Clinical Medicine and Surgery, School of Medicine, University of Naples Federico II, Naples, Italy, **5** Department of Experimental and Clinical Medicine, University of Firenze, Florence, Italy, **6** Department of Health Sciences, University of Catanzaro "Magna Graecia"; Catanzaro, Italy, **7** Rheumatology Unit, Department of Medical and Surgery Sciences, University of Foggia, Foggia, Italy, **8** Rheumatology Unit, Department of Clinical and Experimental Medicine, University of Messina, Messina, Italy, **9** Rheumatic and Systemic Autoimmune Diseases Unit, Department of Biomedical Sciences and Human Oncology, University of Bari Medical School, Bari, Italy, **10** Rheumatology Section, Department of Internal Medicine, University of Palermo, Palermo, Italy

☯ These authors contributed equally to this work.
‡ These authors also contributed equally to this work.
* roberto.giacomelli@cc.univaq.it

## Abstract

### Objective

To assess the predictive role of ferritin and C-reactive protein (CRP) on occurrence of macrophage activation syndrome (MAS) and mortality in patients with adult onset Still's disease (AOSD), a rare and severe disease, included in the multicentre *Gruppo Italiano di Ricerca in Reumatologia Clinica e Sperimentale* (GIRRCS) cohort.

### Methods

The predictive role, at the time of diagnosis, of serum levels of ferritin and CRP on occurrence of MAS and mortality, was evaluated by logistic regression analyses and receiver-operating characteristic (ROC) curves were built to identify patients at high risk of MAS and mortality, respectively.

**Data Availability Statement:** All data generated by the analysis is included into the body of the present work or uploaded as supplementary materials.

**Funding:** The authors received no specific funding for this work.

**Competing interests:** The authors have declared that no competing interests exist.

## Results

In assessed 147 patients with AOSD, levels of ferritin were predictive of MAS (OR: 1.971; P: 0.002; CI 95%: 1.280–3.035). The ROC curve showed that the best cut-off for ferritin was 1225 ng/ml in predicting MAS (sensitivity 88%; specificity 57%). Levels of CRP were predictive of mortality in these patients (OR: 2.155; P: 0.007; CI 95%: 1.228–3.783). The ROC curve showed that the best cut-off for CRP was 68.7 mg/L in predicting mortality (sensitivity 80%; specificity of 65%).

## Conclusions

We reported the predictive role of ferritin and CRP on MAS and mortality, respectively, in a large cohort of patients with AOSD, identifying subsets at higher risk of poor prognosis. Considering that the analysis of CRP and ferritin is widely available, these results could be readily transferable into clinical practice, thus improving the management of patients with AOSD.

## Introduction

Adult-onset Still's disease (AOSD) is a rare and severe inflammatory disease of unknown aetiology, with a higher mortality rate [1]. It usually affects young adults and it is characterised by high spiking fever, arthritis, evanescent rash, and, in more severe cases, by internal organs involvement [2, 3]. Three clinical patterns are generally identified: i. monocyclic pattern, affecting 30% of patients, associated with a single episode and a good prognosis; ii. polycyclic pattern, affecting 30% of patients, characterised by multiple flares alternating with remissions; iii. chronic pattern, affecting 40% of patients, showing a persistently active disease [4]. The latter two patterns are affected by a more severe disease, sometimes requiring hospitalization in intensive care units because of life-threatening complications [5], mostly macrophage activation syndrome (MAS), a reactive form of haemophagocytic lymphohistocytosis (HLH) [6–8]. The treatment of AOSD remains largely empirical, lacking validated guidelines, due to the rarity of the disease [2]. Corticosteroids (CCSs) are considered the first-line therapy, often in combination with synthetic disease-modifying anti-rheumatic drugs (sDMARDs) [9, 10]. In patients, who are inadequate responders to this therapeutic strategy, biological DMARDs (bDMARDs), mainly interleukin (IL)-1 and IL-6 inhibitors, are administered [11–14].

The AOSD inflammatory process may be detected and followed evaluating some specific serum biomarkers. In fact, high levels of C-reactive protein (CRP) are commonly observed during disease flares [1–3]. This is a pentameric protein whose circulating concentrations rise in response to inflammation, following IL-6 secretion by macrophages and T cells [15, 16]. Erythrocyte sedimentation rate (ESR) is also reported to be associated with AOSD disease activity [2]; it is a non-specific measure of inflammation derived by the rate at which red blood cells in anticoagulated whole blood descend in a standardised tube over a period of one hour. Furthermore, a typical 5-fold increase of ferritin levels is reported during AOSD and it is considered a useful marker to assess the activity of the disease [2]. Ferritin is an intracellular iron storage protein [16] and, in addition to its well-known function for the homeostasis of red-blood cells, it is produced and released in the circulation during inflammatory conditions [17]. Possible causes of hyperferritinaemia in AOSD include the enhanced production by macrophages and liver, reduced tissue clearance, and increased production to sequestrate free iron of released haemoglobin due to concomitant erythrophagocytosis [18]. Recently, the concept of

'hyperferritinemic syndrome' has been also suggested, including in a common family AOSD, MAS, catastrophic anti-phospholipid syndrome and septic shock, and hypothesising a ferritin pathogenic role in enhancing the inflammatory burden [18].

Despite many recent progresses in the management of AOSD, the majority of patients may experience recurrent flares, evolving toward the chronic disease pattern and worse prognosis, due to AOSD life-threatening complications [19]. In this context, the need of biomarkers, facilitating early diagnosis and patients profiling is of crucial interest, in identifying those patients with a higher risk of poor outcome [20]. A biomarker generally refers to a measured characteristic which may be used as indicator of some biological state or condition [21], measuring truth, reliability and sensitivity to change, but being also feasible, as suggested by OMERACT filters [22]. Although many biomarkers have been proposed in AOSD, many of these do not met the OMERACT criteria, especially considering the feasibility, which addresses the possibility to be easily and widely used. On the contrary, CRP, ESR, and ferritin are acute-phase reactants, certainly measurable worldwide, also in resource-limited settings, which correlate with the severity of different disorders, including infections and rheumatologic diseases [23]. On these bases, we performed a retrospective analysis of a large cohort of patients (more than 140) with AOSD, to investigate the role of ferritin, ESR, and CRP, at the time of diagnosis, in predicting disease-related mortality and MAS occurrence, during the following prospective follow-up. In addition, we assessed subsets of patients at higher risk of poor prognosis, by the identification of specific cut-offs of these biomarkers.

## Patients and methods

### Study design

The present evaluation has been designed as a retrospective analysis of a prospective followed patients with AOSD, assessed in the multicentre *Gruppo Italiano di Ricerca in Reumatologia Clinica e Sperimentale* (GIRRCS) cohort, to assess the predictive role, at the time of diagnosis, of ferritin, ESR, and CRP on occurrence of complications and mortality. These biomarkers were assessed at the time of diagnosis to perform a prediction in the following prospective follow-up, which is limited to the last available observation of the patients between January 2001 and December 2018. The assessed values were those before that any specific treatment for AOSD was administered. The local Ethics Committee approved the study protocol (*Comitato Etico ASL1 Avezzano-Sulmona-L'Aquila*, *L'Aquila*, Italy, protocol no. 0139815/16) and it has been performed according to the Good Clinical Practice guidelines and the Declaration of Helsinki. After approval of our ethic committee, we collected written informed consents for patients presently and actively followed-up in each centre. However, since the retrospective nature of the study, for those patients who were not anymore followed-up (lost to follow-up or died during the time-period of assessment), after having made every reasonable effort to contact them, we used the fully anonymized clinical data according to the Italian Law on privacy only for research purposes without any other intended aim (*Garante per la protezione dei dati personali*, *Autorizzazione n. 9/2016—Autorizzazione generale al trattamento dei dati personali effettuato per scopi di ricercar scientifica—15 dicembre 2016 [5805552]*). In reporting the results, we followed the STROBE checklist as reported in supplementary material 1 (S1 Table).

### Settings

Patients were selected among those attending Rheumatologic Units of GIRRCS, throughout the Italy. All these units were characterised by experience in management of AOSD as well as in observational studies. Data of participants were recorded during the scheduled visits between January 2001 and December 2018.

## Patients

All patients with AOSD fulfilled the diagnostic criteria proposed by Yamaguchi M. [24]. The assessment at baseline excluded potential mimickers, including infections, cancers, and other autoimmune or autoinflammatory diseases, as previously detailed [19, 25, 26].

## Variables to be assessed

The presence of the following clinical features, at the time of diagnosis, were recorded: fever, typical rash, arthralgia or arthritis, myalgia, lymphadenopathy, sore throat, splenomegaly, hepatomegaly or abnormal liver function tests, abdominal pain, sore throat, weight loss, and gastrointestinal symptoms. The diagnosis of pleural effusion or pleuritis, and lung parenchymal involvement was performed by a chest radiograph or CT scan. After clinical examinations and chest radiographs, patients with clinical suspicion of pericarditis underwent echocardiography. Taking these features together, each patient was also assessed for systemic score [27]. This score assigns 1 point to each of 12 manifestations: fever, typical rash, pleuritis, pneumonia, pericarditis, hepatomegaly or abnormal liver function tests, splenomegaly, lymphadenopathy, leukocytosis > 15000/mm$^3$, sore throat, myalgia, and abdominal pain (max score: 12 points). ESR, ferritin and CRP were also reported, at the time of diagnosis. In addition, during each scheduled examination, each patient was assessed, where appropriate, for the presence of AOSD-related complications including MAS, thrombotic thrombocytopenic purpura, thrombotic microangiopathy, disseminated intravascular coagulopathy, respiratory distress syndrome, diffuse alveolar haemorrhage, pulmonary arterial hypertension, myocarditis, tamponade, constrictive pericarditis, endocarditis, shock, multiple organ failure, fulminant hepatitis, and amyloidosis, as suggested by available literature [28]. MAS diagnosis was defined according to the diagnostic criteria proposed by available literature [29–32]. The therapeutic strategies were also recorded. Treatment used at the time of diagnosis and during follow-up was reported, based on medications administered to each patient for the longest time-period. According to the disease course, at the last scheduled visit, patients were categorised into 4 groups [4]: 3 clinical patterns (monocyclic, polycyclic, chronic) and death. A monocyclic course was defined as a single episode for > 2 months but < 1 year, followed by sustained remission through the whole follow-up. A polycyclic course was characterised by recurrent systemic flares with remission between flares. A chronic course was defined as ≥ 1 episode of persistent symptoms lasting > 1 year. Patients, who died during follow-up, were placed in the fourth group, the AOSD-related death, which was defined as death associated with AOSD or its complications during the follow-up. Remission was defined as the complete disappearance of systemic symptoms and normalisation of laboratory evidence of disease activity for at least 2 consecutive months, regardless of therapy [19, 25, 26]. Flare was characterised by systemic flares occurring after remission. The need of any additional treatment and/or any increased dosage of drugs was considered as a flare of the disease [19, 25, 26].

## Data sources

Relevant data were retrospectively collected by a review of clinical charts, during the scheduled visits for each involved patient. All the data, registered between January 2001 and December 2018, were fully anonymized before we accessed them. Data were collected between January 2019 and June 2019, by a review of clinical charts, which were stored in each of involved centre. All data generated by the analysis is included into the body of the present work or uploaded as supplementary materials.

## Bias

Considering the retrospective design, our study may be subjected to a number of possible biases. We tried to minimise the main methodological problems by a careful definition of each variable to be assessed. Furthermore, patients with significant missing data, which were considered to be meaningful for the analyses, were removed, if one or more missing data in the main outcomes.

## Study size

We aimed at assessing the role of serum biomarkers in on occurrence either of complications or of mortality in patients with AOSD. Considering the rarity of the disease and the retrospective design, no specific sample size was estimated.

## Statistical analysis

The preliminary statistical analysis provided descriptive statistics. Continuous variables which were normally distributed were expressed as mean ± SD, whereas continuous variables which were not normally distributed were expressed as median (interquartile range; IQR). Due to their skewness, ferritin and CRP were ln transformed to purposes of analysis. The correlation analyses among ferritin and CRP with systemic score were performed by Pearson correlation analysis. Logistic regression analyses were performed assessing the statistical significance of possible associations between ferritin, ESR, and CRP on occurrence of MAS and mortality. Since not significant results, ESR was not furtherly assessed in subsequent steps of analyses. The purposeful selection process of covariates started by a univariate analysis of each variable; any variable having a significant univariate test was selected as a possible candidate for the multivariate analysis. Conversely, covariates were removed from the model if not significant. At the end of this multistep process of deleting and refitting, the multivariate models were built, providing OR estimations of significant associations between ferritin and CRP and mortality and MAS occurrence in these patients. In addition, in these multivariate analyses for mortality and MAS occurrence, selected possible clinical confounders (age, gender and CRP or ferritin) were included in addition to positive results retrieved in univariate analyses. Since all the variables that resulted significantly in the univariate analysis were included in the systemic score, because of it is a sum of all the variables tested [27], we added only this value to overcome the limitation of the number of patients with the outcomes interest. In logistic regression output, we included the constant, the value at which the regression line crosses the y-axis (also known as y-intercept), since this ensures that the model will be unbiased (i.e. the mean of the residuals will be exactly zero). After positive findings in logistic regression analyses, receiver-operating characteristic (ROC) curves were performed to evaluate the predictivity of ferritin and CRP in identifying patients with MAS and at high risk of mortality, respectively. The best cut-off for ROC curves was calculated by Youden's index, which is formally calculated as {sensitivity (cut-point) + [specificity (cut-point)– 1]}, and defining the maximum potential effectiveness of our tested biomarkers. Considering the identified cut-offs of ferritin and CRP, we performed further logistic regression analyses assessing the statistical significance of possible relationships among ferritin and CRP and clinical features of patients. Because of the relatively simple design of our study, we had a very low percentage of missing data; thus, very few patients with missing data on main outcomes were removed from the analyses. Statistical significance was expressed by a p value <0.05. The Statistics Package for Social Sciences (SPSS version 17.0, SPSS Inc.) was used for all analyses.

## Results

### Baseline characteristics

Originally, 154 patients were assessed, but 7 patients were not included in the study because of missing values which were meaningful for analysis in the main outcomes (values of ESR, ferritin, CRP, long term follow-up about complications and mortality), thus in the present evaluation, 147 patients were assessed. Demographic and clinical are summarised in Table 1. All these patients showed spiking fever, 88.4% of patients reported arthritis, 74.8% displayed typical skin rash, 61.9% liver involvement, 54.4%. lymphadenopathy, 21.1% pericarditis, and 19.7% pleuritic effusions, respectively. The systemic score resulted to be 6.01±2.01, at the time of diagnosis. At baseline, inflammatory markers resulted increased; median ferritin (IQR)

**Table 1. Demographic and clinical characteristics of assessed patients.**

| Baseline Clinical Characteristics | Assessed Patients N: 147 | Without MAS N: 121 | MAS N: 26 |
|---|---|---|---|
| Age, Mean±SD | 45.2±16.1 | 44.5±15.6 | 48.46±18.4 |
| Female, N (%) | 58 (39.5%) | 50 (41.3%) | 8 (30.7%) |
| **Clinical Features** | | | |
| Fever, N (%) | 147 (100%) | 121 (100%) | 26 (100%) |
| Arthritis, N (%) | 130 (88.4%) | 108 (89.2%) | 22 (84.6%) |
| Skin Rash, N (%) | 110 (74.8%) | 91 (75.2%) | 19 (73.1%) |
| Splenomegaly, N (%) | 98 (66.7%) | 77 (63.6%) | 21 (80.8%) |
| Myalgia, N (%) | 95 (64.6%) | 73(60.3%) | 22 (84.6%) |
| Liver involvement, N (%) | 91 (61.9%) | 70 (57.8%) | 21 (80.8%) |
| Sore throat, N (%) | 83 (56.5%) | 62 (51.2%) | 21 (80.8%) |
| Lymph node, N (%) | 80 (54.4%) | 59 (48.8%) | 21 (80.8%) |
| Pericarditis, N (%) | 31 (21.1%) | 21 (17.4%) | 10 (38.5%) |
| Pleuritis, N (%) | 29 (19.7) | 20 (16.5%) | 9 (34.6%) |
| Abdominal pain, N (%) | 20 (13.6%) | 13(10.7%) | 7 (27.0%) |
| AOSD pneumonia, N (%) | 18 (12.2%) | 14 (11.6%) | 4 (15.4%) |
| Systemic score, Mean±SD | 6.0±2.0 | 5.7±1.9 | 7.6±1.9 |
| **Laboratory Parameters** | | | |
| Leucocytosis >15000mm$^3$, N (%) | 78 (53.1%) | 63 (52.1%) | 15 (46.1%) |
| Ferritin ng/ml, median (IQR) | 1271.0 (2484.0) | 1120.0 (28000.0) | 2746.7 (17136.8) |
| ESR mm/Hr, Mean±SD | 69.4±27.3 | 67.8±27.1 | 76.3±27.8 |
| CRP mg/L, median (IQR) | 50.0 (101.0) | 50.0 (348.0) | 56.5 (204.0) |
| **Therapy** | | | |
| CCSs, N (%) | 147 (100%) | 66 (54.6%) | 20 (76.9%) |
| Low dosage CCSs, N (%) | 61 (41.5%) | 55 (45.4%) | 6 (26.1%) |
| sDMARDs, N (%) | 95 (64.5%) | 75 (62.0%) | 20 (77.0%) |
| bDMARDs, N (%) | 44 (29.9%) | 37 (30.6%) | 7 (27.0%) |
| **Disease course** | | | |
| Monociclic pattern, N (%) | 53 (36.1%) | 49 (40.5%) | 4 (15.4%) |
| Polyciclic pattern, N (%) | 48 (32.7%) | 44 (36.4%) | 4 (15.4%) |
| Chronic pattern, N (%) | 38 (25.9%) | 29 (24.0%) | 9 (34.6%) |
| Mortality, N (%) | 20 (13.6%) | 9 (7.4%) | 11 (42.3%) |
| Follow up, median (IQR) years | 2 (16.4) | 2.5 (14.4) | 1 (16.4) |

AOSD = Adult Onset Still's Disease; CCSs = Corticosteroids; ESR = Erythrocyte Sedimentation Rate; CRP = C Reactive Protein; sDMARDs = synthetic Disease Modifying Anti-Rheumatic Drugs; bDMARDs = biologic Disease Modifying Anti-Rheumatic Drugs; N = Number

1271 (2484) ng/mL; ESR mean±SD 69.35±27.31 mm/hr; median CRP (IQR) 50 (101) mg/L. All the patients were treated with CCSs and the 41.5% of patients were treated with low dosage of CCSs; 64.5% of patients received sDMARDs and 29.9% received bDMARDs. Either at the time of diagnosis or during the follow-up [median 1.2 (range 8) years], we observed that 17.7% of patients were complicated by MAS. Specific characteristics of these patients are shown in Table 1. Finally, we registered a disease-related mortality of 13.6% mainly due to MAS occurrence.

## Correlations among ferritin and CRP with systemic score

The correlation analysis showed that levels of ferritin correlated with systemic score; the Pearson correlation coefficient turned out to be 0.299 (p = 0.0001) pointing out a weak monotonic effect between these 2 variables. In addition, we performed the correlation analysis of ferritin and CRP; the Pearson correlation coefficient resulted to be 0.370 (p = 0.0001). Similarly, a significant weak correlation was retrieved between CRP and systemic score (Pearson correlation coefficient 0.182, p = 0.034). Correlating ESR with these parameters, ESR weakly correlated with systemic score (Pearson correlation coefficient 0.246, p = 0.003) and with CRP (Pearson correlation coefficient 0.359, p = 0.0001), respectively. Conversely, no significant result was obtained correlating ESR with ferritin (Pearson correlation coefficient 0.505 p = 0.065).

## The predictive role of ferritin on MAS in AOSD

As detailed in Table 2, we observed that ferritin, at the time of diagnosis, was predictive of MAS in AOSD. In fact, both univariate analysis (OR: 1.945; P: 0.001; CI 95%: 1.292–2.928) and multivariate analysis (OR: 1.734; P: 0.019; CI 95%: 1.094–2.749) displayed significant results. In multivariate analysis, selected clinical confounders (age, gender and CRP) did not result as being significantly associated with MAS. Additional univariate analyses, about other possible clinical predictors of MAS, were reported in supplementary material 2 (S2 Table). Since all the variables that resulted significantly in the univariate analysis were included in the systemic score, because of it is a sum of all the variables tested [27], we added only this value to overcome the limitation of the number of patients with the outcomes interest. Paralleling with ferritin, the systemic score, at the time of diagnosis, was an independent predictor of MAS (OR: 1.484; P: 0.002; CI 95%: 1.156–1.905). The logistic regression model was statistically significant

**Table 2. Logistic regression analyses assessing ferritin as predictor of MAS.**

| MAS | OR | SE | P | CI 95% |
|---|---|---|---|---|
| | | **Univariate analysis** | | |
| Ferritin | 1.945 | 0.209 | **0.001** | 1.292–2.928 |
| constant | χ2 = 1.641, p = **0.0001** | | | |
| | | **Multivariate analysis** | | |
| Ferritin | 1.734 | 0.235 | **0.019** | 1.094–2.749 |
| Age | 1.012 | 0.015 | 0.442 | 0.982–1.043 |
| Gender | 1.058 | 0.529 | 0.915 | 0.375–2.984 |
| CRP | 0.998 | 0.003 | 0.592 | 0.993–1.004 |
| Systemic score | 1.484 | 0.127 | **0.002** | 1.156–1.905 |
| constant | χ2 = 18.504, p = **0.0001** | | | |

Statistical significance was expressed by a p value <0.05. Bolded values indicate statistically significant results.

AOSD = adult onset Still's disease, CRP = C reactive protein, OR = odds ratio, SE = standard error, P = p-value, CI = confidence interval.

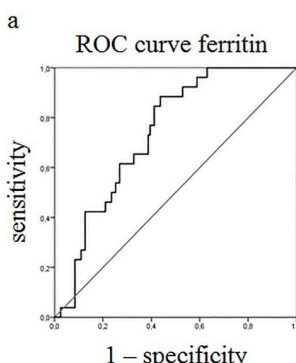 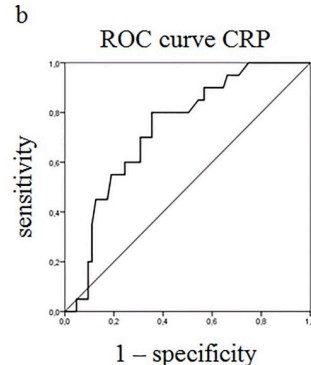

**Fig 1. Receiver operating characteristic (ROC) curves.** a) ROC curve for ferritin and MAS. The area under the ROC curve resulted to be 0.739 (95% CI: 0.652–0.825, p = 0.0001); the analysis showed that the best cut-off for ferritin was 1225 ng/ml in predicting MAS, providing a sensitivity of 88% and a specificity of 57%. b) ROC curve for CRP and mortality. The area under the ROC curve resulted to be 0.735 (95% CI: 0.632–0.838, p = 0.001); the analysis showed that the best cut-off for CRP in predicting mortality was 68.7 mg/L, providing a sensitivity of 80% and a specificity of 65%.

($\chi$2 = 18.504, p = 0.0001). Furthermore, we performed a ROC curve in order to investigate the predictivity of levels of ferritin, at the time of diagnosis, on the likelihood of being diagnosed with MAS (Fig 1). The area under the ROC curve resulted to be 0.739 (95% CI: 0.652–0.825, p = 0.0001) for ferritin. The analysis of ROC curve showed that the best cut-off for ferritin was 1225 ng/ml in predicting MAS, providing a sensitivity of 88% and a specificity of 57%. We also performed a logistic regression analysis to evaluate the possible predictive role of CRP, at the time of diagnosis, on occurrence of MAS, but not significant results were obtained (OR: 1.279; P: 0.199; CI 95%: 0.878–1.862). Similarly, no significant result was retrieved assessing the predictive role of ESR, at the time of diagnosis, on occurrence of MAS (OR: 1.012; P: 0.152; CI 95%: 0.996–1.028).

## Some clinical manifestations as predictors of ferritin $\geq$ 1225 ng/ml

Table 3 displays the contingency table based on the derived cut-off ferritin. Elevation of ferritin was associated with the highest percentage of MAS. Furthermore, ferritin $\geq$ 1225 ng/ml were predictive of some clinical manifestations, including skin rash, AOSD pneumonia, liver involvement and myalgia, as shown in Table 4. The multivariate analysis showed that the evidence of these clinical features was significantly associated with ferritin $\geq$ 1225 ng/ml, skin rash (OR 4.565, 95% CI 1.772–11.761, p = 0.002), AOSD pneumonia (OR 6.712, 95% CI 1.036–43.502, p = 0.046), liver involvement (OR 2.371, 95% CI 1.053–5.339, p = 0.037) and myalgia (OR 2.522, 95% CI 1.122–5.669, p = 0.025). The logistic regression model was

**Table 3. Contingency table for MAS and mortality based on cut-off for ferritin $\geq$ 1225 ng/ml and CRP $\geq$ 68.7 mg/L.**

|  | Ferritin<1225 ng/ml | Ferritin$\geq$1225 ng/ml | Total |
|---|---|---|---|
| Without, N (%) | 64 (54.2%) | 54 (45.8%) | 118 (100%) |
| MAS, N (%) | 3 (11.5%) | 23 (88.5%) | 26 (100%) |
|  | CRP<68.7 mg/L | CRP$\geq$68.7 mg/L | Total |
| No mortality, N (%) | 81 (63.8%) | 46 (36.2%) | 127 (100%) |
| mortality, N (%) | 4 (20.0%) | 16 (80.0%) | 20 (100%) |

MAS = macrophage activation syndrome; CRP = C reactive protein.

**Table 4. Logistic regression analyses assessing clinical manifestations as predictors of ferritin ≥ 1225 ng/ml.**

|  | OR | SE | P | CI 95% |
|---|---|---|---|---|
| **Multivariate analysis** | | | | |
| Skin Rash | 4.565 | 0.483 | **0.002** | 1.772–11.761 |
| Pleuritis | 1.894 | 0.572 | 0.264 | 0.617–5.809 |
| AOSD pneumonia | 6.712 | 0.954 | **0.046** | 1.036–43.502 |
| Liver involvement | 2.371 | 0.414 | **0.037** | 1.053–5.339 |
| Splenomegaly | 1.956 | 0.444 | 0.131 | 0.818–4.674 |
| Lymph node | 1.488 | 0.416 | 0.339 | 0.659–3.360 |
| Myalgia | 2.522 | 0.413 | **0.025** | 1.122–5.669 |
| Constant | $\chi 2$ = 0.669, p = **0.0001** | | | |

Statistical significance was expressed by a p value <0.05. Bolded values indicate statistically significant results.

AOSD = adult onset Still's disease, OR = odds ratio, SE = standard error, P = p-value, CI = confidence interval.

statistically significant ($\chi 2$ = 0.669, p = 0.0001). Univariate analyses of clinical manifestations as predictors of ferritin ≥ 1225 ng/ml were reported in supplementary material 3 (S3 Table).

## The predictive role of CRP on mortality in AOSD

As detailed in Table 5, we observed that CRP, at the time of diagnosis, was predictive of mortality during AOSD. In fact, both univariate analysis (OR: 2.275; P: 0.002; CI 95%: 1.346–3.844) and multivariate analysis (OR: 1.006; P: 0.028; CI 95%: 1.001–1.012) showed that CRP was significantly associated with mortality. In multivariate analysis, selected clinical confounders (age, gender and ferritin) did not result as being significantly associated with mortality. Additional univariate analyses, about clinical predictors of mortality, were reported in supplementary material 3 (S3 Table). Since all the variables that resulted significantly in the univariate analysis were included in the systemic score, because of it is a sum of all the variables tested [27], we added only this value to overcome the limitation of the number of patients with the outcomes interest. Paralleling with CRP, the systemic score, at the time of diagnosis, was an independent predictor of mortality (OR: 1.537; P: 0.003; CI 95%: 1.161–2.034). The logistic regression model was statistically significant ($\chi 2$ = 8.744, p = 0.003). Furthermore, we

**Table 5. Logistic regression analyses assessing CRP as predictor of mortality.**

| MORTALITY | OR | SE | P | CI 95% |
|---|---|---|---|---|
| **Univariate analysis** | | | | |
| CRP | 2.275 | 0.268 | **0.002** | 1.346–3.844 |
| constant | $\chi 2$ = 1.233, p<**0.0001** | | | |
| **Multivariate analysis** | | | | |
| CRP | 1.006 | 0.003 | **0.028** | 1.001–1.012 |
| Age | 1.033 | 0.17 | 0.063 | 0.998–1.067 |
| Gender | 0.677 | 0.581 | 0.502 | 0.217–2.115 |
| Ferritin | 0.913 | 0.264 | 0.731 | 0.544–1.532 |
| Systemic score | 1.537 | 0.143 | **0.003** | 1.161–2.034 |
| constant | $\chi 2$ = 8.744, p = **0.003** | | | |

Statistical significance was expressed by a p value <0.05. Bolded values indicate statistically significant results.

AOSD = adult onset Still's disease, CRP = C reactive protein, OR = odds ratio, SE = standard error, P = p-value,

CI = confidence interval.

**Table 6. Logistic regression analysis assessing clinical manifestations as predictors of CRP ≥ 68.70 mg/L.**

| | OR | SE | P | CI 95% |
|---|---|---|---|---|
| | Multivariate analysis | | | |
| Pericarditis | 1.316 | 1.474 | **0.015** | 1.125–1.800 |
| Lymph node | 1.885 | 0.360 | 0.078 | 0.931–3.814 |
| Myalgia | 3.100 | 0.392 | **0.004** | 1.439–6.681 |
| constant | $\chi 2 = 0.375$, p = **0.0001** | | | |

Statistical significance was expressed by a p value <0.05. Bolded values indicate statistically significant results.
AOSD = adult onset Still's disease, CRP = C reactive protein, OR = odds ratio, SE = standard error, P = p-value,
CI = confidence interval.

performed a ROC curve in order to investigate the predictivity of CRP, at the time of diagnosis, on mortality (Fig 1). The area under the ROC curve resulted to be 0.735 (95% CI: 0.632–0.838, p = 0.001) for CRP. The analysis of ROC curve showed that the best cut-off for CRP in predicting mortality was 68.7 mg/L, providing a sensitivity of 80% and a specificity of 65%. We also performed a logistic regression analysis to evaluate the possible predictive role of ferritin, at the time of diagnosis, on mortality, but not significant results were obtained (OR: 1.318; P: 0.182; CI 95%: 0.879–1.975). Similarly, no significant result was retrieved assessing the predictive role of ESR, at the time of diagnosis, on mortality (OR: 1.014; P: 0.130; CI 95%: 0.996–1.033).

## CRP ≥ 68.7 mg/L as a predictor of some clinical manifestations

Table 3 displays the contingency table based on the derived cut-off of CRP. Elevation of CRP was associated with the highest percentage of mortality. Furthermore, CRP ≥ 68.7 mg/L was a predictor of some clinical manifestations, including pericarditis and myalgia, as shown in Table 6. In multivariate analysis, the evidence of these clinical features was significantly associated with CRP ≥ 68.7 mg/L, pericarditis (OR 1.316, 95% CI 1.125–1.800, p = 0.015) and myalgia (OR 3.100, 95% CI 1.439–6.681, p = 0.004). The logistic regression model was statistically significant ($\chi 2 = 0.375$, p = 0.0001). Univariate analyses of clinical manifestations as predictors of CRP ≥ 68.7 mg/L were reported in supplementary material 5 (S5 Table).

## Discussion

In the era of personalised medicine, the finding of sensitive and specific biomarkers is crucial for improving the clinical practice. Available literature shows an impressive growing number of studies discovering new biomarkers which largely outnumber their possible clinical impacts, especially in rare diseases where the generalisation of the results is limited by low prevalence and incidence. Consequently, new biomarkers may fail in translation to clinical practice because they lack of the necessary sensitivity and specificity to address this clinical need. Thus, we mainly focused our work on widely available biomarkers, ESR, CRP, and ferritin, to be used in rheumatology clinics for prognostication and identification of more severe patients with AOSD. We reported the predictive role of ferritin and CRP, at the time of diagnosis, on occurrence of MAS and mortality, respectively, in a large cohort of patients. We also identified, through the establishment of specific cut-offs, a subset of patients at higher risk of MAS (patients with ferritin more than 1225 ng/ml) and AOSD-related mortality (patients with CRP levels more than 68.7 mg/L). The identification of objective rates, indicative of more aggressive subsets, may help the clinicians to apply additional resources in the care of those patients, at risk of a more severe disease, improving their management. Furthermore, it is also possible to

speculate that these biomarkers, serially performed during the follow-up, may be helpful in monitoring the evolution of the disease allowing clinicians to make the most appropriate decisions.

As far as ferritin is concerned, we tested the hypothesis whether ferritin, at the time of diagnosis, could predict the MAS, providing a useful rationale, readily transferable into clinical practice to alert the physician about the possibility of a patient at higher risk of that complication. Given the presence of HScore and other diagnostic criteria [29–32], we did not aim to provide an additional diagnostic tool, also because a single parameter could not fully assess all clinical features of MAS, but rather a "red flag" for physicians in managing these patients. In our evaluation, we observed that higher levels of ferritin (more than 1225 ng/ml), at the time of diagnosis, were independently predictive of MAS occurrence a higher threshold than what proposed in systemic onset juvenile arthritis [33], which is considered the juvenile counterpart of AOSD [34]. Furthermore, we observed that higher levels of ferritin were predictive of some clinical manifestations, including skin rash, which is associated with a more aggressive course of the disease [35], liver involvement, which may be related to the infiltration of inflammatory cells as well as for the occurrence of haemophagocytosis [36], and AOSD pneumonia. For the latter, considering cost-effectiveness and radiation hazard, a simple blood test would be a good alternative to repeated chest CT for evaluating the current status of lung involvement [37]. It has been suggested that if a biomarker is involved in multiple steps of the disease, its relevance and prognostic values are enhanced [38]. In this context, hyperferritinaemia plays an important role in the differentiation of MAS from other forms of HLH [36, 39]. Furthermore, the expression of ferritin, in bone marrow of patients with AOSD and MAS, correlates with peripheral blood cytopenias and severity [40, 41]. Moreover, AOSD and MAS have been included under the so-called 'hyperferritinemic syndrome' in which the pro-inflammatory proprieties of ferritin could be exacerbated in contributing to the inflammatory burden and development of a cytokine storm [18, 41].

As far as CRP is concerned, paralleling what reported in systemic onset juvenile arthritis [23], we observed that higher CRP, at the time of diagnosis, predicted mortality during the follow-up. This result confirms what observed during infections and other inflammatory diseases, in which CRP correlates with organs failure and poor prognosis in patients admitted to intensive care units [42–44]. CRP is involved in multiple immunoregulatory functions, acting on the complement cascade, binding to C1q, opsonizing bacteria for phagocytosis and stimulating phagocytic cells [45–48]. After its production, the consequent release of pro-inflammatory cytokines, such as IL-1, IL-6 and TNF, may contribute to the development cytokines storm, thus linking the high levels of CRP with the evolution of AOSD. Furthermore, levels of CRP more than 68.7 mg/L predicted some clinical manifestations, including pericarditis and myalgia. Increased CRP levels are common in patients with pericarditis, not only to confirm the suspicion, but also to monitor the disease activity [49]. In addition, different studies showed an association between CRP and muscle pain, although the underlying pathogenic mechanism is not fully elucidated yet [50–52].

Our results showed that both biomarkers displayed a high sensitivity (ferritin 88%, CRP 80%, respectively) associated with a low specificity (ferritin 57%, CRP 65%, respectively). Although it is generally acceptable that markers with high sensitivity may have low specificity, we tried to address these results, in our clinical setting. In our opinion, the following explanations could partially clarify this discrepancy: i. the undefined border between AOSD and MAS, without a clear way point differentiating the two forms; ii. the effects of immunosuppressive therapies, masking the occurrence of MAS in some AOSD patients; iii. the retrospective and multicentre design of this study. During the disease, when the inflammatory process is very pronounced, the border between AOSD and MAS is undefined, with significant overlap

between these two conditions, considered to be included in the same disease spectrum [18, 41]. In fact, less severe or "subclinical" cases of MAS could be masked by immunosuppressive therapies given for a presumed flare of AOSD [53–55], thus possibly limiting the predictivity of our analysis on that. Although future confirmatory studies are needed for the assessment of the validity of these biomarkers, as surrogate endpoints for severity and survival, our results should have significant implications for the care of patients with AOSD and for the design of future clinical trials. In addition, higher CRP and ferritin could reflect higher disease activity, suggesting a more aggressive subset of AOSD. Since the lack of a validated score in AOSD, further studies are necessary to evaluate these biomarkers as reflecting the disease activity in these patients.

In addition, in our cohort, we retrieved that ferritin and CRP paralleled with systemic score in predicting MAS and mortality, respectively. This finding would confirm previous data, showing that systemic score is a clinical tool which could be applied to AOSD, to identify those patients with the likelihood of more severe outcome [6]. In fact, it may suggest that the multi-organ involvement, at the time of diagnosis, is predictive of a poor prognosis [6, 19, 25].

Our study is affected by some limitations. Although searching for biomarkers may improve the management of inflammatory diseases [56, 57], multicentre and retrospective design of this study may be associated with some biases, limiting the external validity of our findings, and reducing the specificity of the prediction of MAS and mortality. Furthermore, recording outcomes of interest after the diagnosis, from few days after that to later in the follow up, according to the data available in clinical charts, could be an additional limitation of the retrospective design of the study. However, it must be pointed out that AOSD is a rare disease, thus designing and performing prospective and adequately powered studies is challenging. On the other side, data deriving from the largest cohorts, in a rare disease must be considered. Moreover, the measurement of these two feasible biomarkers can be easily obtained in the clinical practice, providing relevant prognostic information to the clinicians. Finally, our results would suggest a more rational approach to these patients, avoiding any redundancy in laboratory and instrumental assays, thus facilitating the therapeutic decision-making.

## Conclusions

In conclusion, we reported the predictive role of ferritin and CRP, at the time of diagnosis, on occurrence of MAS and mortality, respectively, in a large cohort of patients with AOSD, identifying subsets at higher risk of MAS (ferritin more than 1225 ng/ml) or mortality (CRP more than 68.7 mg/L). The identification of objective thresholds of biomarkers, suggestive of more aggressive subsets of the disease, could guide the clinicians when to apply additional resources in the care of these patients. Furthermore, after the identification of different clinical subsets, physicians could balance appropriate escalation of therapy, minimising the exposure to iatrogenic harm and avoiding unnecessary expenditures, in patients with less severe disease. Finally, considering that the analysis of CRP and ferritin is widely available, the results of our study may be readily transferable into clinical practice, thus improving the management of patients with AOSD.

## Supporting information

**S1 Table. PRISMA checklist.**
(DOC)

**S2 Table. Univariate logistic regression analyses assessing possible clinical predictors of MAS.**
(DOC)

**S3 Table. Univariate logistic regression analyses assessing possible clinical predictors of ferritin ≥ 1225 ng/ml.**
(DOC)

**S4 Table. Univariate logistic regression analyses assessing possible clinical predictors of mortality.**
(DOC)

**S5 Table. Univariate logistic regression analyses assessing possible predictors of CRP ≥ 68.5 mg/L.**
(DOC)

## Acknowledgments

The authors thank Mrs. Federica Sensini for her technical assistance.

## Author Contributions

**Conceptualization:** Paola Di Benedetto, Paola Cipriani, Ilenia Pantano, Roberto Giacomelli, Piero Ruscitti.

**Data curation:** Paola Di Benedetto, Paola Cipriani, Daniela Iacono, Ilenia Pantano, Francesco Caso, Giacomo Emmi, Rosa Daniela Grembiale, Francesco Paolo Cantatore, Fabiola Atzeni, Federico Perosa, Raffaele Scarpa, Giuliana Guggino, Francesco Ciccia, Roberto Giacomelli, Piero Ruscitti.

**Formal analysis:** Paola Di Benedetto, Paola Cipriani, Daniela Iacono, Ilenia Pantano, Francesco Caso, Giacomo Emmi, Rosa Daniela Grembiale, Francesco Paolo Cantatore, Fabiola Atzeni, Federico Perosa, Raffaele Scarpa, Giuliana Guggino, Francesco Ciccia, Roberto Giacomelli, Piero Ruscitti.

**Investigation:** Paola Di Benedetto, Paola Cipriani, Daniela Iacono, Ilenia Pantano, Francesco Caso, Giacomo Emmi, Rosa Daniela Grembiale, Francesco Paolo Cantatore, Fabiola Atzeni, Federico Perosa, Raffaele Scarpa, Giuliana Guggino, Francesco Ciccia, Roberto Giacomelli, Piero Ruscitti.

**Methodology:** Paola Di Benedetto, Paola Cipriani, Daniela Iacono, Ilenia Pantano, Francesco Caso, Giacomo Emmi, Rosa Daniela Grembiale, Francesco Paolo Cantatore, Fabiola Atzeni, Federico Perosa, Raffaele Scarpa, Giuliana Guggino, Francesco Ciccia, Roberto Giacomelli, Piero Ruscitti.

**Project administration:** Roberto Giacomelli, Piero Ruscitti.

**Resources:** Roberto Giacomelli, Piero Ruscitti.

**Software:** Paola Di Benedetto, Paola Cipriani, Daniela Iacono, Ilenia Pantano, Francesco Caso, Giacomo Emmi, Rosa Daniela Grembiale, Francesco Paolo Cantatore, Fabiola Atzeni, Federico Perosa, Raffaele Scarpa, Giuliana Guggino, Francesco Ciccia, Roberto Giacomelli, Piero Ruscitti.

**Supervision:** Roberto Giacomelli, Piero Ruscitti.

**Validation:** Paola Di Benedetto, Paola Cipriani, Daniela Iacono, Ilenia Pantano, Francesco Caso, Giacomo Emmi, Rosa Daniela Grembiale, Francesco Paolo Cantatore, Fabiola Atzeni, Federico Perosa, Raffaele Scarpa, Giuliana Guggino, Francesco Ciccia, Roberto Giacomelli, Piero Ruscitti.

**Visualization:** Paola Di Benedetto, Paola Cipriani, Daniela Iacono, Ilenia Pantano, Francesco Caso, Giacomo Emmi, Rosa Daniela Grembiale, Francesco Paolo Cantatore, Fabiola Atzeni, Federico Perosa, Raffaele Scarpa, Giuliana Guggino, Francesco Ciccia, Roberto Giacomelli, Piero Ruscitti.

**Writing – original draft:** Paola Di Benedetto, Paola Cipriani, Daniela Iacono, Ilenia Pantano, Francesco Caso, Giacomo Emmi, Rosa Daniela Grembiale, Francesco Paolo Cantatore, Fabiola Atzeni, Federico Perosa, Raffaele Scarpa, Giuliana Guggino, Francesco Ciccia, Roberto Giacomelli, Piero Ruscitti.

**Writing – review & editing:** Paola Di Benedetto, Paola Cipriani, Daniela Iacono, Ilenia Pantano, Francesco Caso, Giacomo Emmi, Rosa Daniela Grembiale, Francesco Paolo Cantatore, Fabiola Atzeni, Federico Perosa, Raffaele Scarpa, Giuliana Guggino, Francesco Ciccia, Roberto Giacomelli, Piero Ruscitti.

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
