## [Decision Letter · Decision Letter 0]

14 May 2020

PONE-D-20-09182

Ferritin and C-reactive protein are predictive biomarkers of mortality and macrophage activation syndrome in adult onset Still’s disease. Analysis of the multicentre Gruppo Italiano di Ricerca in Reumatologia Clinica e Sperimentale (GIRRCS) cohort.

PLOS ONE

Dear Dr. Giacomelli,

Thank you for submitting your manuscript to PLOS ONE. After careful consideration, we feel that it has merit but does not fully meet PLOS ONE’s publication criteria as it currently stands. Therefore, we invite you to submit a revised version of the manuscript that addresses the points raised during the review process.

We would appreciate receiving your revised manuscript by Jun 28 2020 11:59PM. To enhance the reproducibility of your results, we recommend that if applicable you deposit your laboratory protocols in protocols.io, where a protocol can be assigned its own identifier (DOI) such that it can be cited independently in the future. For instructions see: http://journals.plos.org/plosone/s/submission-guidelines#loc-laboratory-protocols

We look forward to receiving your revised manuscript.

Kind regards,

Massimo Cugno, M.D.

Academic Editor

PLOS ONE

Journal Requirements:

2. In the ethics statement in the manuscript and in the online submission form, please provide additional information about the patient records used in your retrospective study, including: a) whether all data were fully anonymized before you accessed them; b) the date range (month and year) during which patients' medical records were accessed; c) the date range (month and year) during which patients whose medical records were selected for this study sought treatment; and d) the source of the medical records analyzed in this work (e.g. hospital, institution or medical center name). If patients provided informed written consent to have data from their medical records used in research, please include this information."

3. "Thank you for including your ethics statement:

'The local Ethics Committee approved the study protocol (ASL1 Avezzano-SulmonaL’Aquila, L’Aquila, Italy protocol no. 0139815/16) and it has been performed according to the Good Clinical Practice guidelines and the Declaration of Helsinki'.

(a) Please amend your current ethics statement to include the full name of the ethics committee/institutional review board(s) that approved your specific study.  

(b) Once you have amended this statement in the Methods section of the manuscript, please add the same text to the “Ethics Statement” field of the submission form (via “Edit Submission”).

Additional Editor Comments (if provided):

The manuscript has been found interesting by all the referees; however, some data underlying the findings are missing in the manuscript and the statistical analysis should be improved. Moreover, according to one of the referees, the revision of the text by a native English speaker is needed. Finally, the manuscript should be assessed by the STROBE checklist (http://www.strobe-statement.org).

Reviewers' comments:

Reviewer's Responses to Questions

**Comments to the Author**

1. Is the manuscript technically sound, and do the data support the conclusions?

Reviewer #1: Yes

Reviewer #2: Partly

Reviewer #3: Yes

2. Has the statistical analysis been performed appropriately and rigorously? 

Reviewer #1: Yes

Reviewer #2: Yes

Reviewer #3: Yes

3. Have the authors made all data underlying the findings in their manuscript fully available?

Reviewer #1: No

Reviewer #2: No

Reviewer #3: Yes

4. Is the manuscript presented in an intelligible fashion and written in standard English?

Reviewer #1: Yes

Reviewer #2: Yes

Reviewer #3: No

5. Review Comments to the Author

Reviewer #1: In their article PD Benedetto et al. reported a strong predictive role of serum ferritin > 1225 ng/ml on occurrence of MAS in a large cohort of well-characterized patients with AOSD, N = 147. Furthermore, serum CRP levels > 68.7 mg/l were predictive of mortality in these patients.

The purpose of this study is clearly defined and, overall, the manuscript is easily understood and well written. The authors stated that the number of patients with missing values was low and that these patients were excluded from the analyses of main outcomes. However, it would be helpful to get the exact numbers of missing values for table 1 as well as for the patients excluded for occurrence of MAS and mortality.

In table 1 and table 3, the subgroup of patients without MAS are given the name “ No MAS”. As “No” is the international abbreviation for “number”, I would suggest to replace it by “WITHOUT MAS”.

Reviewer #2: The authors claim that all the underlying data to perform the analyses mentioned is contained within the manuscript or within the supplemental material. This is not true. More information/data is needed in order to run any of the mentioned logistic regression analyses.

The authors appear to have consulted with a statistician to perform most of the analyses within the manuscript. This is a positive. More, however, needs to be done to exactly explain how the analyses were performed. For instance, the statistical analysis paragraph just mentions that regression analyses were done. The authors should be specific as there are many types of regression analyses. Logistic regression should be stated each time it is used, instead of just regression analyses were done.

The authors also need to do a better job defining their acronyms. OR is used in the text without a definition (odds ratio). Likewise, most tables (and supplemental tables) mention OR,SE,CI, and P without definition. This should be revised.

The text throughout mentions assessing the significance of the multivariate logistic regression model. However, the statistical analysis section does not mention this being done. Please revise to include the method of testing and assessing significance of chi-square value.

The statistical analysis section mentions that there is very little missing data, but that there was some. This does not appear to be mentioned anywhere else in the document. FOr each univariate analysis, please indicate the number of actual samples used in the analysis (especially if it differs from 147). This missing data can become a problem when comparing multivariate models since a subject might not be missing in one model, but is in another. This presents problems for comparing multiple multivariate models, but that is not mentioned here and needs to be considered/dealt with. Additionally, it is not mentioned how the final multivariate model was chosen. The supplemental materials all indicate several variables that were significant at the univariate level that do not seem to have been included at the multivariate level. The text mentions that they might have been included, and then removed if not significant, which is plausible. Howver, the text also seems to indicate that only significant variables should appear in the final multivariate model, and that is not the case. This needs to be better described.

For the correlation analysis paragraph with score, multiple correlations are reported. Each of these is under 0.4, which indicates a weak correlation (although significant). Please amend each instance to say there is a weak correlation. This weak correlation actually allows you to look at multivariate logistic regression with weakly correlated variables, as it does not work well with highly correlated ones.

Supplemental Table1 has a p-value of 2.705. Please correct this value as a p-value > 1 is not possible.

Reviewer #3: The study by Benedetto et al. "Ferritin and CRP are predictive biomarkers of mortality and MAS in AOSD" assessed the predictive value of ferritin and CRP on occurrence of MAS among AOSD patients. The study is interesting and includes a relatively large population of AOSD patients. The results have clinical relevance. The manuscript should be proofreaded.

6. PLOS authors have the option to publish the peer review history of their article (what does this mean?). If published, this will include your full peer review and any attached files.

Reviewer #1: No

Reviewer #2: No

Reviewer #3: No

---

## [Author Response · Author response to Decision Letter 0]

8 Jun 2020

Journal Requirements:

As suggested we revised the text as per PLOS ONE's style requirements. 

2. In the ethics statement in the manuscript and in the online submission form, please provide additional information about the patient records used in your retrospective study, including: a) whether all data were fully anonymized before you accessed them; b) the date range (month and year) during which patients' medical records were accessed; c) the date range (month and year) during which patients whose medical records were selected for this study sought treatment; and d) the source of the medical records analyzed in this work (e.g. hospital, institution or medical center name). If patients provided informed written consent to have data from their medical records used in research, please include this information."

After approval of our ethic committee (Comitato Etico ASL1 Avezzano-Sulmona-L’Aquila, L’Aquila, Italy, protocol no. 0139815/16), we collected written informed consents for patients presently and actively followed-up in each centre. However, since the retrospective nature of the study, for those patients who were not followed-up anymore (lost to follow-up or died during the time-period of assessment), after having made every reasonable effort to contact them, we used the fully anonymized clinical data according to the Italian Law on privacy only for research purposes without any other intended aim (Garante per la protezione dei dati personali, Autorizzazione n. 9/2016 - Autorizzazione generale al trattamento dei dati personali effettuato per scopi di ricerca scientifica - 15 dicembre 2016 [5805552]). Relevant data were retrospectively collected by a review of clinical charts, during the scheduled visits for each involved patient. All the data, collected between January 2001 and December 2018, were fully anonymized before we accessed them. Data were collected between January 2019 and June 2019, by a review of clinical charts, which were stored in each of involved centre. All data, generated by the subsequent analyses, are included into the body of the present work or uploaded as supplementary materials. 

3. "Thank you for including your ethics statement:

'The local Ethics Committee approved the study protocol (ASL1 Avezzano-Sulmona-L’Aquila, L’Aquila, Italy protocol no. 0139815/16) and it has been performed according to the Good Clinical Practice guidelines and the Declaration of Helsinki'.

(a) Please amend your current ethics statement to include the full name of the ethics committee/institutional review board(s) that approved your specific study. 

(b) Once you have amended this statement in the Methods section of the manuscript, please add the same text to the “Ethics Statement” field of the submission form (via “Edit Submission”).

We added that the Comitato Etico ASL1 Avezzano-Sulmona-L’Aquila, L’Aquila, Italy, (protocol no. 0139815/16) approved the study and amended the statement in the submission form. 

Additional Editor Comments (if provided):

The manuscript has been found interesting by all the referees; however, some data underlying the findings are missing in the manuscript and the statistical analysis should be improved. Moreover, according to one of the referees, the revision of the text by a native English speaker is needed. Finally, the manuscript should be assessed by the STROBE checklist (http://www.strobe-statement.org).

Many thanks for the interest in our work, as suggested, we largely revised the English language, we provided the STROBE checklist, we improved the statistical analysis as requested. Concerning the results, all the data are included in the manuscript or in supplementary materials, we better specified this feature. This is a specific statement by a Reviewer 2, but all the variables used to run the logistic analysis are reported in the descriptive tables (table 1, contingency tables, and all supplementary materials). If needed, upon appropriate and motivated request, we can share the dataset used to perform these analyses.

Reviewer #1: 

In their article PD Benedetto et al. reported a strong predictive role of serum ferritin > 1225 ng/ml on occurrence of MAS in a large cohort of well-characterized patients with AOSD, N = 147. Furthermore, serum CRP levels > 68.7 mg/l were predictive of mortality in these patients.

The purpose of this study is clearly defined and, overall, the manuscript is easily understood and well written. The authors stated that the number of patients with missing values was low and that these patients were excluded from the analyses of main outcomes. However, it would be helpful to get the exact numbers of missing values for table 1 as well as for the patients excluded for occurrence of MAS and mortality.

We would like the Reviewer for the interest in our work and apologies for the lack of clarity. We better specified that, originally, 154 patients were assessed, but 7 patients were not included in the study because of missing values that were meaningful for analysis in the main outcomes (values of ESR, ferritin, CRP, long term follow-up about complications and mortality) of our evaluation. Thus, in table 1, there are included only patients without missing data, because those patients with missing data were already excluded. 

In table 1 and table 3, the subgroup of patients without MAS are given the name “ No MAS”. As “No” is the international abbreviation for “number”, I would suggest to replace it by “WITHOUT MAS”.

As suggested, we changed “No MAS” in “Without MAS”. Many thanks for this remark. 

Reviewer #2: 

The authors claim that all the underlying data to perform the analyses mentioned is contained within the manuscript or within the supplemental material. This is not true. More information/data is needed in order to run any of the mentioned logistic regression analyses.

We would like to thank the Reviewer for the interest in our work. All data used to run the logistic regression analyses are included in the manuscript. All the variables are reported in the descriptive tables (table 1, contingency tables, and all supplementary materials). 

The authors appear to have consulted with a statistician to perform most of the analyses within the manuscript. This is a positive. More, however, needs to be done to exactly explain how the analyses were performed. For instance, the statistical analysis paragraph just mentions that regression analyses were done. The authors should be specific as there are many types of regression analyses. Logistic regression should be stated each time it is used, instead of just regression analyses were done.

Many thanks for this remark, we specified that we used a logistic regression analysis every time. 

The authors also need to do a better job defining their acronyms. OR is used in the text without a definition (odds ratio). Likewise, most tables (and supplemental tables) mention OR,SE,CI, and P without definition. This should be revised.

Many thanks for this remark, we specified all the acronyms. 

The text throughout mentions assessing the significance of the multivariate logistic regression model. However, the statistical analysis section does not mention this being done. Please revise to include the method of testing and assessing significance of chi-square value.

As suggested we added in the statistical section the method of testing and assessing the significance of each multivariate logistic regression model. As logistic regression output, we included the constant, the value at which the regression line crosses the y-axis (also known as y-intercept), since this ensures that the model will be unbiased (i.e. the mean of the residuals will be exactly zero). 

The statistical analysis section mentions that there is very little missing data, but that there was some. This does not appear to be mentioned anywhere else in the document. FOr each univariate analysis, please indicate the number of actual samples used in the analysis (especially if it differs from 147). This missing data can become a problem when comparing multivariate models since a subject might not be missing in one model, but is in another. This presents problems for comparing multiple multivariate models, but that is not mentioned here and needs to be considered/dealt with. Additionally, it is not mentioned how the final multivariate model was chosen. The supplemental materials all indicate several variables that were significant at the univariate level that do not seem to have been included at the multivariate level. The text mentions that they might have been included, and then removed if not significant, which is plausible. Howver, the text also seems to indicate that only significant variables should appear in the final multivariate model, and that is not the case. This needs to be better described.

We better specified that, originally, 154 patients were assessed, but 7 patients were not included in the study because of missing values that were meaningful for analysis in the main outcomes (values of ESR, ferritin, CRP, long term follow-up about complications and mortality) of our evaluation. Thus, in all analyses, there are included only patients without missing data, because those patients with missing data were already excluded. Many thanks for the remark, in multivariate analysis, selected clinical confounders (age, gender and CRP or ferritin) were included in addition to positive results of interest retrieved in univariate analyses. Since all the variables, that resulted with a significant value in the univariate analysis, were included in the systemic score (basically, it is a sum of all the variables tested), we added only this value to overcome the limitation of the number of patients with the outcome interest. We better specified these features in the text. 

For the correlation analysis paragraph with score, multiple correlations are reported. Each of these is under 0.4, which indicates a weak correlation (although significant). Please amend each instance to say there is a weak correlation. This weak correlation actually allows you to look at multivariate logistic regression with weakly correlated variables, as it does not work well with highly correlated ones.

As suggested we specified the “weak” correlations according to the value of the coefficient. 

Supplemental Table1 has a p-value of 2.705. Please correct this value as a p-value > 1 is not possible.

Many thanks for this remark, we corrected this mistake. 

Reviewer #3: 

The study by Benedetto et al. "Ferritin and CRP are predictive biomarkers of mortality and MAS in AOSD" assessed the predictive value of ferritin and CRP on occurrence of MAS among AOSD patients. The study is interesting and includes a relatively large population of AOSD patients. The results have clinical relevance. The manuscript should be proofreaded.

We would like to thank the Reviewer for the interest in this work, as suggested we largely revised the English language to improve the readability of our manuscript.

---

## [Editor Report · Decision Letter 1]

15 Jun 2020

Ferritin and C-reactive protein are predictive biomarkers of mortality and macrophage activation syndrome in adult onset Still’s disease. Analysis of the multicentre Gruppo Italiano di Ricerca in Reumatologia Clinica e Sperimentale (GIRRCS) cohort.

PONE-D-20-09182R1

Dear Dr. Giacomelli,

We’re pleased to inform you that your manuscript has been judged scientifically suitable for publication and will be formally accepted for publication once it meets all outstanding technical requirements.

Kind regards,

Massimo Cugno, M.D.

Academic Editor

PLOS ONE
---

## [Editor Report · Acceptance letter]

23 Jun 2020

PONE-D-20-09182R1 

Ferritin and C-reactive protein are predictive biomarkers of mortality and macrophage activation syndrome in adult onset Still’s disease. Analysis of the multicentre Gruppo Italiano di Ricerca in Reumatologia Clinica e Sperimentale (GIRRCS) cohort. 

Dear Dr. Giacomelli:

I'm pleased to inform you that your manuscript has been deemed suitable for publication in PLOS ONE. Congratulations! Your manuscript is now with our production department. 

Kind regards, 

on behalf of

Professor Massimo Cugno 

Academic Editor

PLOS ONE